# KICGPT: Large Language Model with Knowledge in Context for Knowledge Graph Completion

**Yanbin Wei [1, 2], Qiushi Huang[1,3], Yu Zhang[1]\*, James T. Kwok[2]**

[1] Southern University of Science and Technology
[2] Hong Kong University of Science and Technology
[3] University of Surrey

## Abstract

Knowledge Graph Completion (KGC) is crucial for addressing knowledge graph incompleteness and supporting downstream applications. Many models have been proposed for KGC. They can be categorized into two main classes: triple-based and text-based approaches. Triple-based methods struggle with long-tail entities due to limited structural information and imbalanced entity distributions. Text-based methods alleviate this issue but require costly training for language models and specific fine-tuning for knowledge graphs, which limits their efficiency. To alleviate these limitations, in this paper, we propose **KICGPT**, a framework that integrates a large language model (LLM) and a triple-based KGC retriever. It alleviates the long-tail problem without incurring additional training overhead. KICGPT uses an in-context learning strategy called **Knowledge Prompt**, which encodes structural knowledge into demonstrations to guide the LLM. Empirical results on benchmark datasets demonstrate the effectiveness of KICGPT with smaller training overhead and no finetuning.

## 1 Introduction

Knowledge Graphs (KGs) are powerful representations of real-world knowledge. The relationships among entities are captured by triples of the form <head entity, relation, tail entity>. KGs serve as the foundation for various applications such as recommendation systems, question-answering, and knowledge discovery. Knowledge Graph Completion (KGC) plays a crucial role for KGs by completing incomplete triples and hence addressing the inherent incompleteness of KGs. This paper focuses on the link prediction task in KGC, which is to predict the missing entity in an incomplete triple.

Based on the source of information used, existing KGC methods can be categorized into two main classes: triple-based methods and text-based methods (Wang et al., 2022a). Triple-based methods (e.g., TransE (Bordes et al., 2013), R-GCN (Schlichtkrull et al., 2018), and HittER (Chen et al., 2021)) utilize the structure of the knowledge graph as the only source of information for KGC. Given the typical imbalance of entities in KGs, long-tail entities are prevalent and the KGC task has limited structural information about them. Consequently, due to information scarcity, the performance of triple-based methods tends to degrade when processing long-tail entities (Wang et al., 2022a). Text-based methods (such as KG-BERT (Yao et al., 2019)) mitigate this information scarcity problem by encoding textual description as an extra information source (Wang et al., 2022a). Besides the use of pre-trained language models (PLMs), fine-tuning remains necessary for text-based methods to handle diverse knowledge graphs. However, this is resource-intensive and requires task-specific fine-tuning for each downstream task.

Large language models (LLMs), such as Chat-GPT and GPT-4 (OpenAI, 2023), have extensive internal knowledge repositories from their vast pre-training corpora, which can be used as an extra knowledge base to alleviate information scarcity for long-tail entities. Tay et al. (2022), Ouyang et al. (2022) demonstrated the potential for LLMs to handle a wide range of tasks by specially-designed prompts without requiring training or finetuning. Given the appealing properties of LLMs, a straightforward approach is to directly apply LLMs to KGC tasks. However, empirical evidence (Zhu et al., 2023) indicates that pure LLMs still cannot achieve state-of-the-art performance in KGC tasks such as link prediction. Additionally, there are several challenges that hinder the application of LLM on KGC tasks. First, the LLM outputs can be unconstrained and may fall outside the scope of entities in the KGs. Second, LLMs impose length limits on the input tokens, and the limits

---
\*Corresponding author

are far from sufficient for describing a complete KGC task. Lastly, there is no effective in-context learning prompt design for LLM on KGC tasks.

To alleviate the aforementioned limitations, we propose in this paper a **K**nowledge **I**n **C**ontext with **GPT** (**KICGPT**) framework. This integrates LLMs with a traditional structure-aware KG model (which is called a retriever). Specifically, for each query $q = (h, r, ?)$ or $q = (?, r, t)$ in the link prediction task, where "?" denotes the missing tail or head entity to be predicted, the retriever first processes the query $q$ independently and generates an ordered candidate entity list $R_{retriever}$ that ranks all entities in the KG based on their retrieval scores. The LLM then performs re-ranking on the top $m$ entities returned by $R_{retriever}$, and replaces these $m$ entities with re-ranked ones returned by the LLM as the final result $R_{KICGPT}$. To achieve the LLM re-ranking, we propose **Knowledge Prompt**, a strategy of in-context learning (ICL) (Wei et al., 2022), to encode the KG knowledge into demonstrations in prompts. Note that in KICGPT we only need to train the retriever. Compared with existing triple-based methods, KICGPT enables simultaneous utilization of knowledge sources (i.e., KG and LLM's knowledge base) and facilitates their alignment and enrichment to alleviate information scarcity for long-tail entities. Moreover, different from existing text-based models, KICGPT leverages a much larger semantic knowledge base without incurring additional training overhead. Unlike directly applying the LLM to the KGC task, which may generate undesired outputs, the proposed KICGPT constrains its output by formalizing link prediction as a re-ranking task for the given sequence $R_{retriever}$. Moreover, the standard link prediction task requires ranking for all the entities, which is not feasible for LLM due to the length limit on input tokens. To overcome this limitation, KICGPT utilizes the retriever to obtain the top-$m$ entities in $R_{retriever}$, and only allows the LLM to perform re-ranking on these entities.

In summary, our contributions are as follows.

- We propose a novel cost-effective framework **KICGPT** for KGC tasks. To the best of our knowledge, this is the first work that combines LLMs with triple-based KGC methods, offering a unique solution to address the task.

- We propose a novel in-context learning strategy, **Knowledge Prompt**, specifically designed for KGC.

- Extensive experiments on benchmark datasets demonstrate that KICGPT achieves state-of-the-art performance with low training overhead.

## 2 Related Work

**Triple-based KGC**  Most existing KGC methods are triple-based methods, which complete the knowledge graph solely based on the triple information. Early shallow knowledge graph embedding (KGE) methods represent entities and relationships as low-dimensional embedding vectors in a continuous embedding space. Based on the scoring function, these methods can be further categorized (Wang et al., 2017) as translation-based (e.g., TransE (Bordes et al., 2013)) and semantic matching models (e.g., RESCAL (Nickel et al., 2011) and DistMult (Yang et al., 2014)).

However, they suffer from limited expressive power due to the use of shallow network structures. In recent years, more powerful network structures are integrated to solve KGC tasks. Examples include the Graph Neural Networks (Schlichtkrull et al., 2018), Convolutional Neural Networks (Dettmers et al., 2018), and Transformer (Chen et al., 2021). Most of these aggregate local structure context into node embeddings and achieve much improved performance. However, they are still limited by the imbalanced distribution of knowledge graph structure with insufficient knowledge about the long-tail entities. Meta-learning (Xiong et al., 2018; Chen et al., 2019) and logical rules (Sadeghian et al., 2019) can mitigate the long-tail problem in KG. The main difference between them and our work is that they handle long-tail entities by extracting and summarizing common structural patterns or rules from the limited information in KG, while KICGPT combines a vast external knowledge base inside the LLM with the structural information in KGs, which can help alleviate information scarcity.

**Text-based KGC**  In light of the success of Natural Language Processing (NLP), text-based knowledge graph completion is gaining more attention. As another mode of knowledge different from the structured KG, text can provide rich semantic information. DKRL (Xie et al., 2016) first introduces textual descriptions into entity embeddings produced by a convolutional neural network. Subsequent works (such as KG-BERT (Yao et al., 2019), KEPLER (Wang et al., 2021b), and Pretrain-KGE

(Zhang et al., 2020b)) use a pre-trained language model (PLM) to encode the text descriptions. More recently, LMKE (Wang et al., 2022a) proposed a contrastive learning framework that uses the PLM to obtain entity and relation embeddings in the same space as word tokens, and demonstrated its effectiveness on long-tail problem These methods generally rely on the language models to process text descriptions and require finetuning for different knowledge graphs. The proposed KICGPT, which uses LLM directly, is more efficient because it is training-free and requires no finetuning.

**LLMs for KGs** Some recent works also explore the ability of LLM on KG tasks. StructGPT (Jiang et al., 2023) proposed a general framework for improving zero-shot reasoning ability of LLMs on structured data. It leverages the LLM to perform reasoning on the KG Question Answering (KGQA) task with the help of auxiliary interfaces that fetch the needed information from KG. Though both StructGPT and our work utilize KG and LLM, StructGPT aims to help the LLM to handle structured data and explore KGQA tasks, while ours utilizes the knowledge base inside the LLM to handle KGC's long-tail problem. Moreover, StructGPT performs multi-step reasoning directly based on the KG structure, while the proposed KICGPT utilizes the KG information in a different way. First, we use the whole KG to generate preliminary results before using the LLM. Moreover, we incorporate a portion of the KG triples into ICL demonstrations to help the LLM conduct reasoning. Zhu et al. (2023) directly evaluated the performance of a LLM on KG reasoning. Our work design an ICL strategy to guide the LLM to perform reasoning. Moreover, their experiments on the link prediction task only involve 25 sampled instances from the FB15k-237 dataset (which has 20,466 test triples). Instead, we re-run their setting on ChatGPT and report the new results as a baseline in our experiments.

**ICL for LLMs** A typical way of approaching LLM is through in-context learning (Brown et al., 2020), by providing explicit instructions and demonstrations to guide the model's behavior. This approach has been effective in various language understanding and generation tasks (Rae et al., 2021; Wei et al., 2022). In-context learning exposes the model to specific contextual information, allowing it to grasp and reproduce necessary patterns and

structures for precise generation (Ouyang et al., 2022). However, the success of in-context learning depends heavily on the quality of the prompt, and crafting suitable prompts can be delicate (Wang et al., 2022b). While there has been work exploring ICL on different tasks (Dong et al., 2022), to the best of our knowledge, no such work has been done for KGC. Unlike existing works, the proposed ICL strategy, Knowledge Prompt, considers the characteristics of KGC tasks and demonstrates its effectiveness on KGC tasks.

## 3 Methodology

In this section, we introduce the proposed KICGPT model. The complete algorithm is in appendix A.1.

### 3.1 Problem Setting

A knowledge graph can be represented as a set of triples $G = \{(h, r, t)\}$, where $E$ and $R$ denote the set of entities and relations in $G$, respectively, $h \in E$ is a head entity, $t \in E$ is a tail entity, and $r \in R$ represents the relation between them. Link prediction is an important task in KGC. Given an incomplete triple $(h, r, ?)$ or $(?, r, t)$ as query, link prediction aims to predict the missing entity (denoted as $?$). A link prediction model usually needs to score all plausible entities as missing entity and then rank all entities in descending order. For simplicity of presentation, we focus on queries with missing tail entities (i.e., $(h, r, ?)$). Queries with missing head entities (i.e., $(?, r, t)$) can be handled analogously.

### 3.2 Overview

Figure 1 illustrates the KICGPT framework. There are two components: a triple-based KGC retriever and a LLM. For each query triple $(h, r, ?)$, the retriever first generates the score of $(h, r, e)$ for each entity $e \in E$. The ranking (in descending score) of all the entities is denoted $R_{retriever} = [e_{1st}, e_{2nd}, e_{3rd}, \ldots, e_{|E|th}]$. The LLM then performs a re-ranking of the top-$m$ entities based on its knowledge and demonstrations in the proposed Knowledge Prompt. The re-ranking $R_{LLM} = [e'_{1st}, e'_{2nd}, e'_{3rd}, \ldots, e'_{mth}]$ is a permutation of $[e_{1st}, e_{2nd}, e_{3rd}, \ldots, e_{mth}]$. By replacing the $m$ leading entities in $R_{retriever}$ by $R_{LLM}$, the KICGPT outputs $R_{KICGPT} = [e'_{1st}, e'_{2nd}, e'_{3rd}, \ldots, e'_{mth}, e_{m+1th}, \ldots, e_{|E|th}]$.

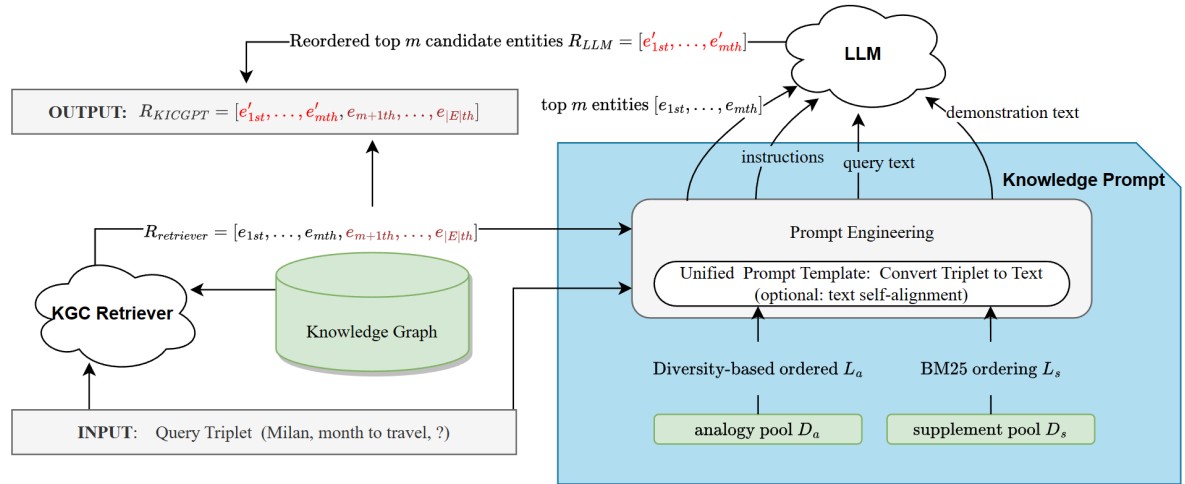

Figure 1: An illustration of the KICGPT framework.

## 3.3 Knowledge Prompt

In this section, we introduce Knowledge Prompt, an in-context learning strategy specially designed for KGC tasks. By encoding part of the KG into demonstrations, Knowledge Prompt boosts the performance of LLM on link prediction.

### 3.3.1 Demonstration Pool

For each query $(h, r, ?)$, we construct two pools of triples, $D_a$ and $D_s$, from the KG as demonstrations.

**Analogy Pool** The analogy pool $D_a$ contains triples that help the LLM to better understand the semantics of the query by analogy. Let $G_{train}$ and $G_{valid}$ be the sets of KG triples for training and validation, respectively. $D_a = \{(e', r, e'') \in G_{train} \cup G_{valid}|e', e'' \in E\}$ includes triples with the same relation as the query $(h, r, ?)$.

**Supplement Pool** The supplement pool $D_s$ contains triples which provide supplementary information about the query's head entity $h$. Specifically, $D_s$ includes all triples with $h$ as the head or tail entity in the training and validation parts: $\{(h, r', e') \in G_{train} \cup G_{valid}|r' \in R, e' \in E\} \cup \{(e', r', h) \in G_{train} \cup G_{valid}|r' \in R, e' \in E\}$.

### 3.3.2 Demonstration Ordering

As the demonstration order affects performance (Ye et al., 2023), we propose different ordering strategies for the analogy and supplement pools.

For the analogy pool $D_a$, all its triples are similar to the query because they share the same relation. As diversity of demonstrations is important so that the LLM can learn from various analogy demonstrations, we propose an ordering strategy

that promotes diversity. Specifically, first we set a zero counter for each entity. A triple from $D_a$ is randomly selected as demonstration, and the counters associated with its entities are increased by 1. We iteratively choose as demonstration the triple whose entities' associated counters have the smallest sum (tie is resolved randomly). The associated counters are then increased by 1. This is repeated until all triples in $D_a$ are used, and the resultant demonstration list obtained is denoted $L_a$.

For the supplement pool $D_s$, as it serves to provide supplementary information on the query's head entity $h$, we prefer related demonstrations to the query. Specifically, we rank all triples in $D_s$ according to their BM25 scores (Robertson and Walker, 1994). This score is used to evaluate the correlation between texts in each demonstration and query. The resultant ranking list, denoted $L_s$, contains all triples in $D_s$.

Queries with a specific relation or head entity use a shared analogy or supplement pool. To prevent double counting, demonstration pools are created and ordered in data pre-processing. During inference, the corresponding pool is used based on the query.

### 3.3.3 Prompt Engineering

Prompt engineering is an important in ICL (Zhao et al., 2021). In this section, we show the whole interaction workflow with the LLM, and also considerations of the prompt design.

The demonstrations and query take the form of triples. However, LLMs require natural language input. To remedy this gap, KICGPT uses a unified prompt template to convert the query and

demonstrations to plain text with the same format. We then perform multi-round interactions with the LLM, so as to guide it to perform re-ranking, where these texts and some instructions are organized as prompt inputs. Figure 2 illustrates the workflow of a multi-round interaction with the LLM. For each link prediction query $(h, r, ?)$, KICGPT creates an independent conversational prompt. The whole multi-round interaction process includes four stages: responsibility description, question and demonstration description, multiple demonstrations, and final query.

Responsibility description is illustrated in the first part of Figure 2. We tell the LLM that its role is an assistant to ranking candidate answers for a question based on plausibility. We then check its feedback to ensure that it knows the task.

The question and demonstration description stage is shown in the second part of Figure 2. We input the question (corresponding to the query's text) and tell it that two types of examples are to be provided and should be treated differently, one for analogy and one containing supplementary information.

Next, illustrated in the third part of Figure 2, we provide the LLM with a batch of demonstrations ($L_a$ and $L_s$) from the analogy pool and supplement pool. To include more KG information and demonstrations, we repeat this step as many times as possible subject to the input token length limit.

Finally, we restate the query text, and ask the LLM to re-rank the top-$m$ candidate entities. The feedback is parsed into an ordered list $R_{LLM}$, which then replaces the top-$m$ entities in $R_{retriever}$ as the final answer.

### 3.4 Text Self-Alignment

In this section, we propose Text Self-alignment for KG text cleaning. It transforms the raw text in the KG to more understandable descriptions by the LLM.

Raw and obscure text descriptions generally exist in the KG. For example, a raw relation description in the FB15k-237 dataset is *"/tv/tv_program/country_of_origin"*. Most existing methods (Yao et al., 2019; Zhang et al., 2020b; Wang et al., 2021b, 2022a) convert it to a cleaner description by removing the symbols (e.g., *"tv tv program country of origin"*). By default, KICGPT uses "of" to organize such hierarchical relation, leading to *"country of origin of tv program of tv"*.

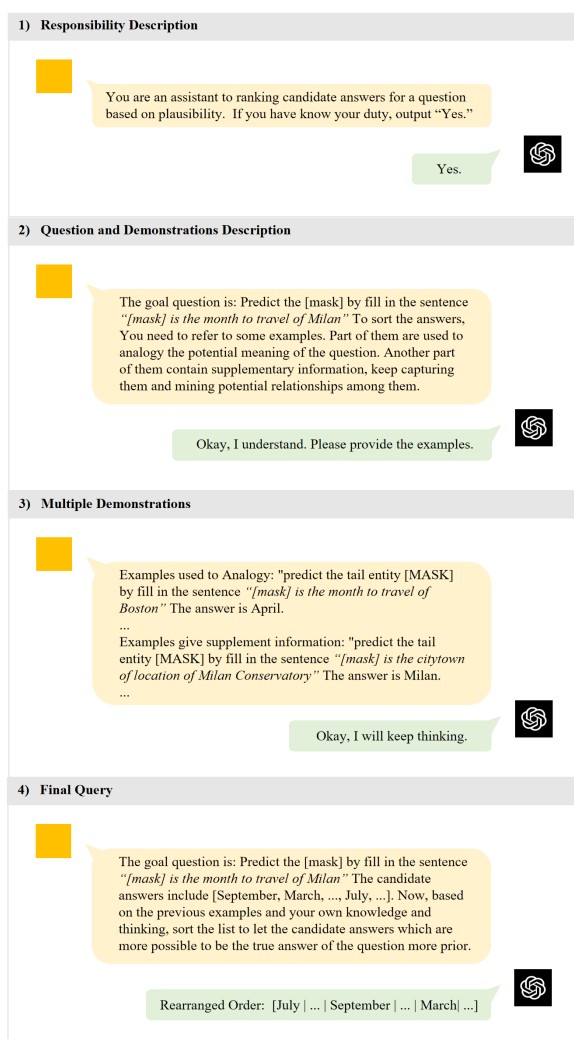

Figure 2: Illustration of a multi-round interaction with the LLM. Stage 3 is repeated many times to provide more demonstrations.

However, this description may still be hard for the LLM to comprehend, and may lead to incorrect responses.

To address the above issue, we use ICL and let the LLM to summarize natural text descriptions from the given demonstrations that use raw descriptions. Specifically, for a relation $r$, we use the analogy pool $D_a$ for $r$ (Section 3.3.1) as demonstrations and order them to get the ordered list $L_a$ (Section 3.3.2). These demonstrations are fed as a prompt to the LLM (Appendix A.2), which is asked to summarize the semantics of the relation into a sentence (the self-aligned text description of $r$). For the above example with raw relation text *"/tv/tv_program/country_of_origin"*, the LLM generates the much clearer and natural self-aligned text description *"[T] is the country where the TV*

| Dataset | FB15k-237 | | | | WN18RR | | | |
|---|---|---|---|---|---|---|---|---|
| Metric | MRR | Hits@1 | Hits@3 | Hits@10 | MRR | Hits@1 | Hits@3 | Hits@10 |
| **Triple-based methods** | | | | | | | | |
| RESCAL(Nickel et al., 2011) ♣ | 0.356 | 0.266 | 0.390 | 0.535 | 0.467 | 0.439 | 0.478 | 0.516 |
| TransE (Bordes et al., 2013) ♠ | 0.279 | 0.198 | 0.376 | 0.441 | 0.243 | 0.043 | 0.441 | 0.532 |
| DistMult (Yang et al., 2014) ♠ | 0.241 | 0.155 | 0.263 | 0.419 | 0.430 | 0.390 | 0.440 | 0.490 |
| ComplEx (Trouillon et al., 2016) ♠ | 0.247 | 0.158 | 0.275 | 0.428 | 0.440 | 0.410 | 0.460 | 0.510 |
| RotatE (Sun et al., 2019) | 0.338 | 0.241 | 0.375 | 0.533 | 0.476 | 0.428 | 0.492 | 0.571 |
| TuckER (Balažević et al., 2019) | 0.358 | 0.266 | 0.394 | 0.544 | 0.470 | 0.443 | 0.482 | 0.526 |
| HAKE (Zhang et al., 2020a) | 0.346 | 0.250 | 0.381 | 0.542 | 0.497 | 0.452 | 0.516 | 0.582 |
| CompGCN (Vashishth et al., 2019) | 0.355 | 0.264 | 0.390 | 0.535 | 0.479 | 0.443 | 0.494 | 0.546 |
| HittER(Chen et al., 2021) | 0.344 | 0.246 | 0.380 | 0.535 | 0.496 | 0.449 | 0.514 | 0.586 |
| **Text-based methods** | | | | | | | | |
| Pretrain-KGE (Zhang et al., 2020b) | 0.332 | - | - | 0.529 | 0.235 | - | - | 0.557 |
| KG-BERT (Yao et al., 2019) ♠ | - | - | - | 0.420 | 0.216 | 0.041 | 0.302 | 0.524 |
| StAR (Wang et al., 2021a) ♠ | 0.263 | 0.171 | 0.287 | 0.452 | 0.364 | 0.222 | 0.436 | 0.647 |
| MEM-KGC (w/o EP) (Choi et al., 2021) | 0.339 | 0.249 | 0.372 | 0.522 | 0.533 | 0.473 | 0.570 | 0.636 |
| MEM-KGC (w/ EP) (Choi et al., 2021) | 0.346 | 0.253 | 0.381 | 0.531 | 0.557 | 0.475 | 0.604 | **0.704** |
| **LLM-based methods** | | | | | | | | |
| ChatGPT$_{zero-shot}$ (Zhu et al., 2023) ◇ | - | 0.237 | - | - | - | 0.190 | - | - |
| ChatGPT$_{one-shot}$ (Zhu et al., 2023) ◇ | - | 0.267 | - | - | - | 0.212 | - | - |
| KICGPT | **0.412** | **0.327** | **0.448** | 0.554 | 0.549 | 0.474 | 0.585 | 0.641 |
| KICGPT$_{tsa}$ | 0.410 | 0.321 | 0.430 | **0.581** | **0.564** | **0.478** | **0.612** | 0.677 |

Table 1: Comparison between the proposed methods and baseline methods. The best result in terms of each metric is shown in **bold** and the second best one is underlined. ♠ indicates that results are copied from (Wang et al., 2022a), ♣ implies that results are copied from (Chen et al., 2021), ◇ means that the results are running on the entire data according to the settings in (Zhu et al., 2023), and other results are taken from their original papers. EP denotes the entity prediction task in the MEM-KGC model.

*program* $[H]$ *originated from."*, where $[T]$ and $[H]$ are placeholders for the tail and head entities, respectively.

Data pre-processing uses text self-alignment to create clear and aligned descriptions for each relation, which can be directly used as relation text in KICGPT for link prediction. This variant is called KICGPT$_{tsa}$. Since the texts are derived from the LLM, they conform to its presentation conventions, making them easier to understand and improving performance.

## 4 Experiment

In this section, we empirically evaluate KICGPT.

### 4.1 Setup

**Datasets.** We evaluate the proposed methods on the FB15k-237 (Toutanova et al., 2015) and WN18RR (Dettmers et al., 2018), which are widely-used benchmark for link prediction. FB15k-237 is a subset of the freebase (Bollacker et al., 2008) knowledge graph, which includes commonsense knowledge about topics such as movies, sports, awards, and traveling. WN18RR is a subset of WordNet (Miller, 1995), which contains knowledge about English morphology. Both the FB15k-237

and WN18RR datasets remove redundant inverse relations in case of information leakage. Compared with FB15k-237, the knowledge graph of WN18RR is much sparser. The statistics of the two datasets are shown in Table 2.

| Dataset | # Ent | # Rel | # train | # valid | # test |
|---|---|---|---|---|---|
| WN18RR | 40,943 | 11 | 86,835 | 3,034 | 3,134 |
| FB15k-237 | 14,541 | 237 | 272,115 | 17,535 | 20,466 |

Table 2: Statistics of the benchmark datasets.

**Baselines.** We compare the proposed KICGPT with a number of triple-based, text-based and LLM-based baselines. The triple-based baselines include RESCAL (Nickel et al., 2011), TransE (Bordes et al., 2013), DistMult (Yang et al., 2014), ComplEx (Trouillon et al., 2016), RotatE (Sun et al., 2019), TuckER (Balažević et al., 2019), HAKE (Zhang et al., 2020a), CompGCN (Vashishth et al., 2019), and HittER (Chen et al., 2021). The text-based baselines[1] include Pretrain-KGE (Zhang et al., 2020b), KG-BERT (Yao et al., 2019), StAR (Wang et al., 2021a), and MEM-KGC (Choi et al.,

---
[1]We did not compare our method with Wang et al. (2022a) due to a data leakage issue on the entity degrees.

2021). The LLM-based baselines are based on ChatGPT. Zhu et al. (2023) reported zero-shot and one-shot link prediction results on 25 instances sampled from FB15k-237. We run their codes on the whole dataset and report the updated results as ChatGPT$_{zero-shot}$ and ChatGPT$_{one-shot}$.

**Implementation details.** Though the proposed method can be used with various retrievers and LLMs, we prefer lightweight models for efficiency considerations. Specifically, our KICGPT implementation uses RotatE (Sun et al., 2019) as the retriever and ChatGPT (gpt-3.5-turbo) as the LLM. Hyper-parameters for RotatE are set as in (Sun et al., 2019). The value of $m$ is selected from {10, 20, 30, 40, 50}, and the batch size from {4, 8, 16, 32} on a small random subset with 200 instances from the validation set. For the ChatGPT API, we set temperature, presence_penalty, and frequency_penalty to 0, and top_p to 1 to avoid randomness. The detailed prompts are shown in Appendix A.2.

**Metrics.** Link prediction outputs a ranked list of all KG entities. We report Mean Reciprocal Rank (MRR), and Hits@1, 3, 10 of the ranked list under the "filtered" setting. The "filtered" setting (Bordes et al., 2013) is a common practice that filters out valid entities other than ground truth target entities from the ranked list.

### 4.2 Experimental Results

Results are shown in Table 1.[2] As can be seen, the proposed KICGPT and KICGPT$_{tsa}$ achieve state-of-the-art performance on both FB15k-237 and WN18RR in most metrics.

Specifically, on the Fb15k-237 dataset, both variants of KICGPT surpass all the baselines in terms of all the evaluation metrics. The text self-alignment method helps improve the Hits@10 performance but degrades the MRR and Hits@{1,3} performance. This may be because these demonstrations and aligned text are mutually supportive, which enhances the confidence of LLM to understand correct semantics, but for relations that LLM does not understand well, the aligned texts may not convey the true semantics.

On the Fb15k-237 dataset, both KICGPT and KICGPT$_{tsa}$ surpass all the baselines in terms of all evaluation metrics. Text self-alignment helps

improve the Hits@10 performance but degrades the MRR and Hits@{1,3} performance. This may be because the text self-alignment texts are derived from the demonstrations, which are also used in the LLM inference process along with the text self-alignment texts. While the mutually supportive text self-alignment texts and demonstrations enhance the LLM's understanding of correct semantics, they also make incorrect interpretations of relational semantics more persistent.

| WN18RR | FB15k-237 |
|---|---|
| _also_see | /tv/tv_program/country_of_origin |
| _hypernym | /location/location/partially_contains |
| _has_part | /common/topic/webpage. /common/webpage/category |

Table 3: Example relations from the WN18RR and FB15k-237 datasets.

On the WN18RR dataset, KICGPT$_{tsa}$ achieves state-of-the-art performance on all metrics except Hits@10. Unlike the FB15k-237 dataset, text self-alignment improves all metrics. This may be partly because the higher average number of triples per relation in WN18RR (8454.8) compared to FB15k-237 (1308.5), which facilitates the generation of more precise text descriptions due to the increased availability of information for each relation. Besides, as illustrated in Table 3, WN18RR relations have more concise and direct semantics than those in FB15k-237, making text self-alignment for WN18RR easier than for FB15k-237. These differences cause the LLM to summarize more concise and precise descriptions for relations from WN18RR, which boosts the performance with higher-quality aligned text.

The proposed methods show better performance over the triple-based and text-based baselines. This demonstrates the usefulness of the knowledge base inside the LLM. Besides, compared with the LLM-based baselines (i.e., ChatGPT baselines[3]), the proposed methods significantly outperform on both datasets, which shows the superiority of integrating KG with LLMs for KGC. The performance

---

[2]We only report Hits@1 for the LLM-based baselines because the LLM only outputs an answer but not a list.

[3]The results we report for ChatGPT baselines are worse than those reported in (Zhu et al., 2023). This is because they experimented on only 25 triples on the FB15k-237 dataset while we test on the whole 20,466 testing triples. Besides, they manually restated descriptions in KG as clean and natural expressions for the 25 triple, but doing this for all the testing triples is labor-intensive and time-costly. In our experiment, we use the text generated by text self-alignment as a substitution.

improvement mainly comes from the injection of information contained in the knowledge graph.

The demonstration ordering process takes 50.13 and 30.29 minutes for the FB15k-237 and WN18RR datasets, respectively. In terms of training efficiency, KICGPT takes only 28.36 and 20.63 minutes to train on FB15k-237 and WN18RR, respectively, making it more efficient compared to existing text-based methods that can take hours to train. This is because KICGPT does not require fine-tuning of the LLM.

## 4.3 Ablation Study

In this experiment, using the FB15k-237 dataset, we perform ablation studies to demonstrate usefulness of each component in KICGPT. Table 4 shows the results.

In the first ablation study, we shuffle the top-$m$ entities in $R_{retriever}$ before feeding into the LLM. As can be seen, this causes a slight performance degradation. This shows that the order offered by the retriever is important and can reduce the difficulty of re-ranking. Recently, Sun et al. (2023) also noted the significance of the initial order in search re-ranking.

In the second study, we randomize the demonstration orders from the analogy and supplement pools. Again, the performance degrades, demonstrating effectiveness of the proposed demonstration ordering in Section 3.3.2.

In the third study, we retrieve random triples from the KG as demonstrations. The resultant performance degradation shows usefulness of the construction of demonstration pools in Section 3.3.1.

In the fourth study, we exclude all demonstrations, while still providing the top-$m$ candidates to constrain the ChatGPT output. As can be seen, a significant performance gap with the full model is observed, demonstrating the necessity of the proposed ICL strategy. Note also that Hit@3 without ICL is superior to Hit@3 with random demonstrations. This may be attributed to the vast number of triples in the KG, rendering the random demonstrations to have insignificant semantic relevance to the query. The use of irrelevant triples as demonstrations to LLM can introduce noise, potentially resulting in misleading outcomes.

In the last ablation study (Appendix A.2), we use a prompt without the prompt engineering component (Section 3.3.3). This variant still uses the proposed demonstration pools (Section 3.3.1) and

ordering (Section 3.3.2). The observed degraded performance implies that the specially designed prompts can make use of the properties of the KGC task to boost performance.

|  | MRR | Hits@1 | Hits@3 | Hits@10 |
|---|---|---|---|---|
| KICGPT | **0.412** | **0.327** | **0.448** | **0.554** |
| shuffle candidates | 0.401 | 0.312 | 0.433 | 0.521 |
| w/o demonstration ordering | 0.368 | 0.283 | 0.417 | 0.497 |
| random demonstrations | 0.349 | 0.271 | 0.387 | 0.481 |
| w/o ICL | 0.342 | 0.241 | 0.403 | 0.481 |
| w/o prompt engineering | 0.401 | 0.307 | 0.432 | 0.548 |

Table 4: Ablation results on **FB15k-237** (averaged over 4 random runs).

## 4.4 Analysis on Long-Tail Entities

To show the effectiveness of KICGPT and KICGPT$_{tsa}$ to handle long-tail entities, we follow (Wang et al., 2022a) and group entities by their logarithm degrees in the knowledge graph. Entities with lower degrees (and hence more likely to be long-tail entities) are assigned to groups with lower indexes. A triple $(h, r, t)$ is considered relevant to group $d$ if $h$ or $t$ belongs to $d$.

Figure 3 displays the Hits@1 and Hits@10 performance averages of various models on the FB15k-237 dataset, categorized by the logarithm of entity degrees. Text-based methods demonstrate slightly better performance than triple-based methods on long-tail entries. However, this improvement is not significant since it only applies to a small portion of long-tail entities (specifically, groups 0, 1, 2). Compared with these baselines, the proposed models achieve performance improvement on almost all groups and perform significantly better on long-tail entities, which confirms the benefits of combining the LLM with KG.

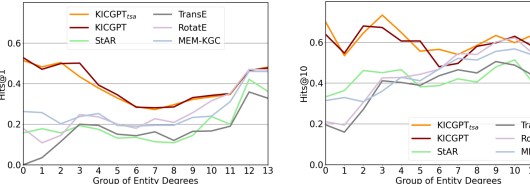

Figure 3: Average performance of different models (in terms of Hits@1 and Hits@10) grouped by the logarithm of entity degrees on the FB15k-237 dataset.

## 5 Conclusion

In this paper, we propose KICGPT, an effective framework to integrate LLM and traditional KGC methods for link prediction, and a new ICL strategy

called Knowledge Prompt. KICGPT utilizes the LLM as an extra knowledge base. Compared with text-based methods, KICGPT utilizes the training-free property of LLM to significantly reduce training overhead, and does not require finetuning for different KGs. Experimental results show that KICGPT achieves state-of-the-art performance on the link prediction benchmarks and is effective in the handling of long-tail entities.

## Acknowledgements

This work is supported by NSFC general grant 62076118 and Shenzhen fundamental research program JCYJ20210324105000003.

## Limitations

The performance improvement of the proposed approach in KGC is largely due to the large and extensive knowledge base within the LLM. However, for some KGs (such as personal preference data for users in an E-commerce platform), the LLM may not contain enough relevant knowledge. Therefore, the proposed method works mostly for common-sense KGs. Besides, because of the limited token length in the LLM, we cannot inject all relevant facts from the KG as prompts.

## Ethics Statement

There is no ethical problem in our study.

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

# A Appendix

## A.1 KICGPT Algorithm

We provide the algorithm of KICGPT here.

---

**Algorithm 1:** Demonstration pools from KG

---

**Input:** KG $G = (h, r, t)$, $E$, and $R$ denote the entity set and the relation set of $G$, $G_{train}$ and
$G_{valid}$ are sets of triples in $G$ for training and validation
link prediction query $q = (h, r, ?)$

**Output:** analogy pool $D_a$
supplement pool $D_s$

1  $D_a = \{(e', r, e'') \in G_{train} \cup G_{valid} | e', e'' \in E\}$
2  $D_s = \{(h, r', e') \in G_{train} \cup G_{valid} | r' \in R, e' \in E\} \cup \{(e', r', h) \in G_{train} \cup G_{valid} | r' \in R, e' \in E\}$

---

---

**Algorithm 2:** Diversity-based ordering for analogy pool

---

**Input:** Analogy demonstration pool: $D_a$, entity set $E$ of KG

**Output:** ordered list $L_a$

1  **foreach** *element $e \in E$* **do**
2  $\quad$ initialize a zero counter $c_e = 0$ for $e$
3  Randomly select a triple $(h', r', t')$ from $D_a$, append it to $L_a$ and increase the counter $c_{h'}$ and $c_{t'}$ by 1
4  **repeat**
5  $\quad$ Find a triple $a$ that has the minimum sum of $c_{e'}$ and $c_{e''}$ where $e'$ and $e''$ are the head and tail entity in triple $a$ (If there are multiple triples with minimum sum of entity counters, randomly select one of them as $a$)
6  $\quad$ Append $a$ to $L_a$ and remove $a$ from $D_a$
7  $\quad$ increase the counter of entities $c_{e'}, c_{e''}$ in triple $a$ by 1
8  **until** *$D_a$ is empty*

---

---

**Algorithm 3:** BM25 score-based ordering for supplement pool

---

**Input:** Supplement demonstration pool: $D_s$
        link prediction query $q$
**Output:** ordered list $L_s$

1 **foreach** *triple s in $D_s$* **do**
    calculate the BM25 score between text descriptions of $s$ and $q$
2 $L_s \leftarrow$ Ordering triples in $D_s$ according to the BM25 score in descending order

---

---

**Algorithm 4:** Algorithm for KICGPT

---

```
/* For presentation simplicity, we represent the algorithm on missing tail queries. missing
   head queries are handled in a similar way.                                          */
```

**Input:** KG $G = (h, r, t)$, $E$, and $R$ denote the entity set and the relation set of $G$, $G_{train}$ and
        $G_{valid}$ are sets of triples in $G$ for training and validation
        link prediction query $q = (h, r, ?)$
**Output:** Ranked list $R_{KICGPT}$ contains all $e \in E$

1 Ranked list $R_{retriever}$ contains all $e \in E \leftarrow$ Handling $q$ by triple-based KGC retriever
    $R_{retriever} = [e_{1st}, e_{2nd}, e_{3rd}, \ldots, e_{|E|th}]$
2 Analogy Pool $D_a$, Supplement Pool $D_s \leftarrow$ Fetch Demonstration Pools from G based on
    $q = (h, r, ?)$
3 $L_a \leftarrow$ Ordering triples in $D_a$ based on diversity
4 $L_s \leftarrow$ Ordering triples in $D_s$ based on BM25 score
5 Use a unified prompt template to convert the query $q$ and triples in $L_a$ and $L_s$ to text
6 With some instructions, demonstrations arranged by order in $L_a$ and $L_s$ are organized to ICL
    prompts
7 Given the ICL prompts and query, LLM is asked to perform re-ranking for first m entities in
    $R_{retriever}$
    $R_{LLM} \leftarrow$ LLM (first m entities in $R_{retriever}$, ICL prompts, text of $q$)
    $R_{LLM} = [e'_{1st}, e'_{2nd}, e'_{3rd}, \ldots, e'_{mth}]$, is permutation of first m entities in $R_{retriever}$
8 $R_{KICGPT} = [e'_{1st}, e'_{2nd}, e'_{3rd}, \ldots, e'_{mth}, e_{m+1th}, \ldots, e_{|E|th}]$, where first m entities is $R_{LLM}$ and
    remaining come from corresponding location of $R_{retriever}$

---

## A.2 Prompt Formatting

We list the prompts used in this paper as follows.

| Datasets | Prompt Template | ChatGPT |
|----------|-----------------|---------|
| FB15k-237 | You are a good assistant to reading, understanding and summarizing. Ecuador is the partially_contains of location of location of Pacific Ocean
Appalachian Mountains is the partially_contains of location of location of Massachusetts
Moldavia is the partially_contains of location of location of Moldova
Ecuador is the partially_contains of location of location of South America
Adirondack Mountains is the partially_contains of location of location of Warren County
In above examples, What do you think "partially_contains of location of location of " mean? Summarize and descript its meaning using the format: "If the example shows something A is partially_contains of location of location of of something B, it means A is [mask] of B." Fill the mask and the statement should be as short as possible. | If the example shows something A is partially_contains of location of location of of something B, it means A is located partially within the boundaries of B. |
| WN18RR | You are a good assistant to reading, understanding and summarizing. red indian be member of domain usage of disparagement compass be member of domain usage of archaism
penicillin v potassium be member of domain usage of trade name
nanna be member of domain usage of united kingdom of great britain and northern ireland
nerves be member of domain usage of plural form
zyloprim be member of domain usage of trademark
In above examples, What do you think "be member of domain usage of" mean? Summarize and descript its meaning using the format: "If the example shows something A be member of domain usage of something B, it means A is [mask] of B." Fill the mask and the statement should be as short as possible. | If the example shows something A be member of domain usage of something B, it means A is a term or word that belongs to the category or domain of B's usage. |

Table 5: Examples of prompts for text self-alignment.

| Prompt Formatting | Prompt Template | ChatGPT |
|---|---|---|
| Trivial Prompt | predict the tail entity [MASK] from the given (Mel Blanc,type_of_union of marriage of people of spouse_s of person of people of , [MASK]) by completing the sentence "what is the type_of_union of marriage of people of spouse_s of person of people of Mel Blanc? The answer is ". The answer is Marriage, so the [MASK] is Marriage. predict the tail entity [MASK] from the given (Screen Actors Guild Life Achievement Award,award_winner of award_honor of award of winners of award_category of award of , [MASK]) by completing the sentence "what is the award_winner of award_honor of award of winners of award_category of award of Screen Actors Guild Life Achievement Award? The answer is ". The answer is Stan Laurel, so the [MASK] is Stan Laurel.
The list of candidate answers is [Marriage, Domestic partnership, Civil union, Official Website, Rang De Basanti, HBO, Male, Television, Judaism-GB, Crusades]. And the question is predict the tail entity [MASK] from the given (Stan Laurel,type_of_union of marriage of people of spouse_s of person of people of , [MASK]) by completing the sentence "what is the type_of_union of marriage of people of spouse_s of person of people of Stan Laurel? The answer is ". Now, based on the previous examples and your own knowledge and thinking, sort the list to let the candidate answers which are more possible to be the true answer to the question prior. Output the sorted order of candidate answers using the format "[most possible answer \| second possible answer \| ... \| least possible answer]" and please start your response with "The final order:". Do not output anything except the final order. Note your output sorted order should contain all the candidates in the list but not add new answers to it. | The final order: [Marriage \| Domestic partnership \| Civil union \| Official Website \| HBO \| Male \| Television \| Rang De Basanti \| Judaism-GB \| Crusades]. |

Table 6: Trivial Prompts on the FB15k-237 dataset.

| Steps | Prompt Template | ChatGPT |
|---|---|---|
| Responsibility Description | You are a good assistant to perform link prediction and sorting. Given a goal question and a list of candidate answers to this question. You need to order these candidate answers in the list to let candidate answers which are more possible to be the answer to the question prior. If you have known your responsibility, respond "Yes". Otherwise, respond "No". Do not output anything except "Yes" and "No". | Yes. |
| Question and Demonstration Description | The goal question is: predict the tail entity [MASK] from the given (Stan Laurel,type_of_union of marriage of people of spouse_s of person of people of, [MASK]) by completing the sentence "what is the type_of_union of the marriage of people of spouse_s of the person of people of Stan Laurel? The answer is ". To sort the candidate answers, typically you would need to refer to some other examples that may be similar to or related to the question. Part of the given examples are similar to the goal question, you should analogy them to understand the potential meaning of the goal question. Another part of the given facts contains supplementary information, keep capturing this extra information and mining potential relationships among them to help the sorting. Please carefully read, realize, and think about these examples. Summarize the way of thinking in these examples and memorize the information you think maybe help your sorting task. During I give examples please keep silent until I let you output. | Okay, I understand. I will wait for your examples and instructions. |
| Multiple Demonstrations | Examples used to Analogy: "predict the tail entity [MASK] from the given (Mel Blanc,type_of_union of marriage of people of spouse_s of person of people of , [MASK]) by completing the sentence "what is the type_of_union of marriage of people of spouse_s of person of people of Mel Blanc? The answer is ". The answer is Marriage, so the [MASK] is Marriage."

Examples give supplement information: "predict the tail entity [MASK] from the given (Screen Actors Guild Life Achievement Award,award_winner of award_honor of award of winners of award_category of award of , [MASK]) by completing the sentence "what is the award_winner of award_honor of award of winners of award_category of award of Screen Actors Guild Life Achievement Award? The answer is ". The answer is Stan Laurel, so the [MASK] is Stan Laurel. Keep thinking but not output. | Okay, I will keep thinking and analyzing the given examples to identify potential relationships and patterns that can help with the sorting task. |
| Final Query | The list of candidate answers is [Marriage, Domestic partnership, Civil union, Official Website, Rang De Basanti, HBO, Male, Television, Judaism-GB, Crusades]. And the question is predict the tail entity [MASK] from the given (Stan Laurel,type_of_union of marriage of people of spouse_s of person of people of , [MASK]) by completing the sentence "what is the type_of_union of marriage of people of spouse_s of person of people of Stan Laurel? The answer is ". Now, based on the previous examples and your own knowledge and thinking, sort the list to let the candidate answers which are more possible to be the true answer to the question prior. Output the sorted order of candidate answers using the format "[most possible answer | second possible answer | ... | least possible answer]" and please start your response with "The final order:". Do not output anything except the final order. Note your output sorted order should contain all the candidates in the list but not add new answers to it. | The final order: [Marriage \| Domestic partnership \| Civil union \| Official Website \| HBO \| Male \| Television \| Rang De Basanti \| Judaism-GB \| Crusades]. |

Table 7: Examples of prompts for KICGPT on the FB15k-237 dataset.

| Steps | Prompt Template | ChatGPT |
|-------|----------------|---------|
| Responsibility Description | Assume you're a linguist of English lexicons. You will be first given some examples. Then use these examples as references and your own knowledge to score for some statements. If you have known your responsibility, respond "Yes". Otherwise, respond "No". Do not output anything except "Yes" and "No". | Yes. |
| Question and Demonstration Description | The goal statements are about member of domain usage of trade name. trade name : a name given to a product or service. Part of the given examples are similar to the statements, you should analogy them to understand the potential meaning of the statements to be scored. Another part of the given examples contains supplementary information, keep capturing this extra information and mining potential relationships among them to help the scoring. Please carefully read, realize and think about these examples. Summarize the way of thinking in these examples and memorize the information you think maybe help. DO NOT give me any feedback. | Okay. |
| Multiple Demonstrations | Examples used to Analogy: trade name : a name given to a product or service. vinblastine : periwinkle plant derivative used as an antineoplastic drug (trade name Velban) that disrupts cell division. vinblastine be member of domain usage of trade name . vernacular : a characteristic language of a particular group (as among thieves); "they don't speak our lingo". chink : a narrow opening as e.g. between planks in a wall. chink be member of domain usage of vernacular.
Examples give supplement information: trade name : a name given to a product or service. cortone acetate : a corticosteroid hormone (trade name Cortone Acetate) normally produced by the adrenal cortex; is converted to hydrocortisone. cortone acetate be member of domain usage of trade name . trade name : a name given to a product or service. phenelzine : monoamine oxidase inhibitor (trade name Nardil) used to treat clinical depression. phenelzine be member of domain usage of trade name. Keep thinking but DO NOT give me any feedback. | Okay. |
| Final Query | trade name : a name given to a product or service. verapamil : a drug (trade names Calan and Isoptin) used as an oral or parenteral calcium blocker in cases of hypertension or congestive heart failure or angina or migraine. verapamil be member of domain usage of trade name. Directly give a score out of 100 for the statement and DO NOT output any other thing
...
trade name : a name given to a product or service. nitrostat : trade names for nitroglycerin used as a coronary vasodilator in the treatment of angina pectoris. nitrostat be member of domain usage of trade name .. Directly give a score out of 100 for the statement and DO NOT output any other thing
...
trade name : a name given to a product or service. hydantoin : any of a group of anticonvulsant drugs used in treating epilepsy. hydantoin be member of domain usage of trade name .. Directly give a score out of 100 for the statement and DO NOT output any other thing.
... | 90.
...
100
...
50
... |

Table 8: Examples of prompts for KICGPT on the WN18RR dataset.