# OpenReview forum: "KICGPT: Large Language Model with Knowledge in Context for Knowledge Graph Completion"
_EMNLP/2023/Conference — EMNLP 2023 Findings_

### Official Review · Reviewer_dRov · 2023-08-07

**Soundness:** 4

**Excitement:**

3: Ambivalent: It has merits (e.g., it reports state-of-the-art results, the idea is nice), but there are key weaknesses (e.g., it describes incremental work), and it can significantly benefit from another round of revision. However, I won't object to accepting it if my co-reviewers champion it.

**Paper Topic And Main Contributions:**

The paper proposes a framework for coupling a large language model with a triple-based retriever for knowledge graph completion (i.e. triple completion). The authors introduce a strategy for including relevant triple-demonstrations in a multi-prompt interaction with ChatGPT along with a prompt-engineering methodology for verbalising relation texts into fluent textual description using larger, OpenAI models. They compare their approach against a variety of triple- and text-based baselines (incl. ChatGPT on a zero-shot and one-shot setup) on two different triple completion benchmarks: FB15k-237 and WN18RR.

**Questions For The Authors:**

Question A: Section 3.3.3 describes a multi-round prompting strategy for re-ranking the resulting entities. Were the answers from ChatGPT consistent, throughout the various prompting turns, across all test cases in the two datasets? How are cases during which the OpenAI model would return an un-expected output at one or more dialogue turns are handled?
Question B: I understand from the manuscript that the results from ChatGPT are not fully-conditioned to the entities that are returned by the retriever. Did the output from ChatGPT include only permutations of the original set of candidate entities, across all test instances?
Question C: While the baseline solutions appear to improve their performance as the group of entity degrees is increasing (i.e. in Figure 3), a similar behaviour is not observed for the KICGPT systems. This is more evident for the Hits@1 case. The manuscript could benefit from some additional clarification about why this behaviour is observed.

**Reasons To Accept:**

- The paper is nicely written; the ideas and the train-of-thought of the experiments are clearly described.
- Interesting method for coupling triple-based approaches for triple completion with larger language models.
- Promising idea for including and ordering demonstrations of different nature (i.e. coming from an analogy and a supplement pool). The ordering aspect can be also particularly useful for cases when the length of the input sequence should be truncated.

**Reasons To Reject:**

- The work is partially motivated by the challenges faced by triple-based approaches when dealing with long-tail entities. However, the manuscript does not highlight any particular considerations for long-tail triples, and I understand that any performance improvements for such cases (presented in Section 4.4) can be mostly attributed to ChatGPT (i.e. assuming that long-tail entities would lack suitable demonstration examples in their corresponding supplement pool).
- Some details about how the experiments are conducted are missing (please refer to Question A and B below).
- RotatE is used as the retriever in the proposed architecture. However, this choice is not sufficiently justified in the manuscript. I believe it would be interesting to see the performance of the end system, using at least one more from the KG embedding models.

**Reproducibility:**

3: Could reproduce the results with some difficulty. The settings of parameters are underspecified or subjectively determined; the training/evaluation data are not widely available.

**Reviewer Confidence:**

4: Quite sure. I tried to check the important points carefully. It's unlikely, though conceivable, that I missed something that should affect my ratings.

**Typos Grammar Style And Presentation Improvements:**

- I believe the dataset labels (i.e. FB15k-237 and WN18RR) in Table 2 should be swapped.

---

> ### Author Rebuttal · Authors · 2023-08-29
>
> Thanks for your constructive comments. The typo about the dataset labels in Table 2 will be corrected in the revision.  We address your concerns as follows.
>
> > ${\bf Q1}$: Section 3.3.3 describes a multi-round prompting strategy for re-ranking the resulting entities. Were the answers from ChatGPT consistent,
> throughout the various prompting turns, across all test cases in the two datasets? How are cases
> during which the OpenAI model would return an unexpected output at one or more dialogue turns are handled?
>
> ${\bf Answer}$: As shown by the examples in Appendix A.2, the instructions usually do very well in limiting the LLM output to a given set of candidate answers and help keep the consistency of the LLM output in each round.
>
> However, there are still some rare cases that need to be handled.
>
>
> 1. Outputting entities beyond the candidate set: In this case, we directly filter out the entities that are not in the candidate set.
>
> 2. Losing some candidate entities: In this case, we append the lost entities according to their corresponding initial ordering (by retriever) at the end of LLM output. Note that this case typically occurs when the LLM is very confident about the top answers and it tends to omit the others.
>
> 3. Damaged output: In this case, we cannot identify a valid ordering from the LLM output. To handle this, KICGPT directly uses the initial ordering of the retriever as the final decision (i.e., it degenerates to the retriever).
>
> These rare cases are very rare. For example, in fb15k-237, such rare cases account for only 2.26\% (926 queries out of 40932). For Wn18RR, we do not have rare cases because we explicitly let the LLM to score  and
> rank
> the entities.
>
> > ${\bf Q2}$: I understand from the manuscript that the results from ChatGPT are not fully-conditioned to the entities that are returned by the retriever. Did the output from ChatGPT include only permutations of the original set of candidate entities, across all test instances?
>
> ${\bf Answer}$: Please refer to the answer to the previous question.
>
> >${\bf Q3}$: While the baseline solutions appear to improve their performance as the group of entity degrees is increasing (i.e. in Figure 3), a similar behaviour is not observed for the KICGPT systems. This is more evident for the Hits@1 case. The manuscript could benefit from some additional clarification about why this behaviour is observed.
>
> ${\bf Answer}$: Thanks for your insightful question. Unlike most baselines that show a performance increase with an increase in degree, the performance of KICGPT in Figure 3 follows a distinct pattern. It initially peaks on long-tail entities, then decreases with an increase in degree, and finally reaches another peak on popular entities. This can be explained as follows:
>
> For the long-tail entities, very few entities are directly related to them, and retrievers can easily include all these entities in the candidate set. Moreover, we observe that the answers to long-tail queries are highly likely to be included in the union of the analogy pool and candidate set. Therefore, because the true answers are far more reasonable than the other entities, the LLM can use its powerful capacity to find them out easily. This corresponds to the first performance peak in Figure 3.
>
> For the middle-range entities (which fall between popular and long-tail entities), we observed that relatively more adversarial or confusing candidates are mixed with the true answers in the initial ordering. Identifying the true answers becomes more difficult in this case. Besides, compared with the long-tail and popular entities, we observed relatively more cases where the true answer is not in the candidate set. This results in a decrease in performance, explaining why KICGPT does not follow the pattern seen in baseline solutions.
>
> For the popular entities in KG, the initial ordering is more reasonable. It places the true answers at the top of the list and confusing candidates at the back, which reduces the difficulty of LLM reordering. This ultimately results in another performance peak in Figure 3.
>
> We believe that this clarification will improve our manuscript and we will include a more detailed discussion in the revision.
>
> > ${\bf Q4}$: I understand that any performance improvements for such cases (presented in Section 4.4) can be mostly attributed to ChatGPT.
>
> ${\bf Answer}$: The performance improvement on long-tail entities is partially due to the external knowledge in LLM. However, we believe that the KG structure also plays an important role in performance improvement.
>
> One contribution the KG structure offers is a set of candidate entities to LLM for answering queries. This not only restricts the LLM from generating unexpected output beyond the KG, but also provides the LLM the most likely candidate answers to the query.
>
> Another contribution is to provide a good initial ordering of candidate entities,
>  which can alleviate the challenge of reordering. This is consistent with a recent study that employed LLM in reordering tasks [1].
>
>  The third contribution is that, even in the absence of supplement demonstrations for long-tail entities, the analogy pool can still aid LLMs in comprehending the semantics of long-tail queries.
>
>
>
> The following table shows the performance of a pure ChatGPT (without the proposed KICGPT components on long-tail entities) on groups with different degrees on the FB15k-237 dataset. Note that this pure ChatGPT also has demonstrations, which are randomly selected from the original KG and use the same prompt template as KICGPT. However, the pure ChatGPT is asked to provide the answer directly, without the aid of the ordered candidate answers. As can be seen, KICGPT significantly outperforms pure ChatGPT even on long-tail entities. Therefore, we believe that the performance improvement on long-tail entities is not solely dependent on the LLM. All the components in KICGPT (including the retriever (ordered candidates) and Knowledge Prompt (well-selected, ordered, and prompt-organized demonstrations)) also play critical roles in achieving state-of-the-art performance.
>
> |  Model | degree 0 | degree 1 | degree 2-3 | degree 4-7 |
> |---|---|---|---|---|
> | KICGPT | 0.5278 | 0.4706 | 0.4988 | 0.5009 |
> | ChatGPT | 0.3721 | 0.3357 | 0.3662 | 0.3479 |
>
> Hit@1 for long-tail performance on FB15k-237
>
> [1]  Is ChatGPT Good at Search?
> Investigating Large Language Models as Re-Ranking Agent, arXiv:2304.09542, 2023.
>
> > ${\bf Q5}$: RotatE is used as the retriever in the proposed architecture. However, this choice is not sufficiently justified in the manuscript. I believe it would be interesting to see the performance of the end system, using at least one more from the KG embedding models.
>
>
> ${\bf Answer}$: Thank you for your valuable suggestion. In order to choose the most suitable retriever for our model, we experimented with several lightweight methods, including TransE, RotatE, Distmult, and ComplEX, on a randomly sampled tiny dataset consisting of 200 instances from the validation set. As can be seen from the table below, RotatE is very effective.
>
> | Metric | RotatE | TransE | DistMult | ComplEx |
> |---|---|---|---|---|
>  | Hit@1 | 0.341 | 0.292 | 0.265 | 0.279 |
>  | Hit@10 | 0.610 | 0.583 | 0.516 | 0.544 |
>
> KICGPT performance for different retrievers on the sampled tiny dataset
>
> To further support our decision, we plan to include experiment results based on another KG embedding method in the revised version of our paper.
>
> Sincerely thanks again for your patient and valuable review. If you have any other questions or advice, please do not hesitate to contact us.

---

### Official Review · Reviewer_A54b · 2023-08-07

**Soundness:** 3

**Excitement:**

3: Ambivalent: It has merits (e.g., it reports state-of-the-art results, the idea is nice), but there are key weaknesses (e.g., it describes incremental work), and it can significantly benefit from another round of revision. However, I won't object to accepting it if my co-reviewers champion it.

**Paper Topic And Main Contributions:**

This paper tackles a problem of knowledge graph completion (KGC) and proposes an approach (KICGPT) that integrates in-context learning and a triple-based KGC retriever. Concretely, this paper focuses on the link prediction task where a model is asked to predict either a head entity or a tail entity given an incomplete triple (i.e., (head, relation, ?) or (?, relation, tail)). First, a triple-based KGC retriever ranks candidate entities. Subsequently, an LLM is used as a re-ranker with curated prompts (e.g., different types of demonstrations and their order). The re-ranked entities are obtained by four rounds of interaction with the LLM. Additionally, the text self-alignment step  can be conducted if a raw text in KG is not LLM friendly (i.e., not natural sentences).

This approach is evaluated on the KGC benchmarks such as FB15k-235 and WN18RR and compared with both text-based and triple-based approaches. In terms of MRR and Hits@k, this approach (w and w/o text self-alignment) outperforms all prior approaches on both benchmarks.


**Reasons To Accept:**

- Using an LLM as a re-ranker for KGC is an interesting idea, and this paper provides a study of prompt selection strategies.
- The proposed approach surpasses prior work in two benchmarks, demonstrating the effectiveness of this method.
- This paper clearly articulates the key ideas and the motivations behind them. Both the design choices and experimental setup are sound and reasonable.


**Reasons To Reject:**

- In the experiments, this approach was implemented using a proprietary language model (ChatGPT). The prompt engineering method relies heavily on the behavior of the LM, so it might not generalize well to other LLMs.
- The ablation study reveals that the improvement over the trivial prompt is modest, raising questions about its effectiveness relative to the cost.


**Reproducibility:**

3: Could reproduce the results with some difficulty. The settings of parameters are underspecified or subjectively determined; the training/evaluation data are not widely available.

**Reviewer Confidence:**

3: Pretty sure, but there's a chance I missed something. Although I have a good feel for this area in general, I did not carefully check the paper's details, e.g., the math, experimental design, or novelty.

---

> ### Author Rebuttal · Authors · 2023-08-29
>
> Thanks for your constructive comments. We address your concerns as follows.
>
> > ${\bf Q1}$: The prompt engineering method
> relies heavily on the behavior of the LM, so it might not generalize well to other
> LLMs.
>
>
> ${\bf Answer}$: While we agree that the choice of LLM is important, our model is agnostic to the specific LLM used.
>
> The LLM in our approach serves as a black box for the re-ranking task, regulated by specific prompt instructions (as detailed in Appendix A.2). Our model's variable-length and truncated prompt design (Section 3.3.3) allows it to accommodate existing LLMs with varying token length limits.
>
> To further validate our model's adaptability, we plan to extend our experiments to include open-source LLMs like Llama-2. This will allow us to explore the use of visible parameters or embeddings in conjunction with Knowledge Graph structures.
>
>
> > ${\bf Q2}$: The ablation study reveals that the improvement over the trivial prompt
> is modest, raising questions about its effectiveness relative to the cost.
>
> ${\bf Answer}$: To clarify, in the ablation experiments, the "Trivial prompt" variant only removes the prompt engineering component (Section 3.3.3) from KICGPT. As such, this variant still benefits from our special demonstration pools (Section 3.3.1) and ordering methods (Section 3.3.2), which are also supported by the initial order provided by the retriever. The performance gap observed in this variant is solely due to the absence of prompt engineering, which incurs a negligible cost (i.e., adding prompt engineering of Section 3.3.3). To prevent confusion, we will rename this variant to "Without prompt engineering" in the revised version.
>
> Sincerely thanks again for your valuable review. If you have any other questions, please do not hesitate to contact us.

---

### Official Review · Reviewer_C1He · 2023-08-10

**Soundness:** 4

**Excitement:**

4: Strong: This paper deepens the understanding of some phenomenon or lowers the barriers to an existing research direction.

**Paper Topic And Main Contributions:**

The paper points out the significance of Knowledge Graph Completion (KGC) in addressing incomplete triples within Knowledge Graphs (KGs). Specifically, this paper centers its focus on the link prediction task within KGC for incomplete KGs. It identifies limitations inherent in two existing KGC methods, prompting the need for an alternative approach.
In response to these limitations, the paper introduces KICGPT—a novel approach that leverages in-context learning (ICL) through a strategy termed Knowledge Prompt. Notably, KICGPT showcases promising results in predicting missing entities within KGs. The authors assert that their approach is noteworthy for being the first to synergize Large Language Models (LLMs) with triple-based KGC methods, thereby achieving state-of-the-art performance with minimal training overhead.

**Reasons To Accept:**

The paper points out there are novel approaches contribution which I think can benefit the community. Even though there are many existing KGC triple-based methods. However, the KICGPT "combines a vast external knowledge base inside LLM with the structural information in KGs", which is a unique solution to address the KGC tasks. Also, they specially designed a strategy for KGC tasks called Knowledge Prompt, which further improve the performance.

**Reasons To Reject:**

I believe the paper is well-structured and has valuable contributions. However, there are a few instances where the concept of incomplete triple representation is repeated in both the introduction and problem setting sections.

**Reproducibility:**

4: Could mostly reproduce the results, but there may be some variation because of sample variance or minor variations in their interpretation of the protocol or method.

**Reviewer Confidence:**

3: Pretty sure, but there's a chance I missed something. Although I have a good feel for this area in general, I did not carefully check the paper's details, e.g., the math, experimental design, or novelty.

---

> ### Author Rebuttal · Authors · 2023-08-29
>
> Thanks for your thoughtful comments.
>
> As you said, we will revise the paper to avoid redundancy in the final revision.
>
> Sincerely thanks for your encouraging rating and valuable review. If you have any other questions, please do not hesitate to contact us.

---

### Official Review · Reviewer_QYzB · 2023-08-11

**Typos Grammar Style And Presentation Improvements:** 1. There are some wrong words in the …
**Soundness:** 3

**Excitement:**

3: Ambivalent: It has merits (e.g., it reports state-of-the-art results, the idea is nice), but there are key weaknesses (e.g., it describes incremental work), and it can significantly benefit from another round of revision. However, I won't object to accepting it if my co-reviewers champion it.

**Missing References:**

None

**Paper Topic And Main Contributions:**

This paper proposes to integrate a LLM and a triple-based KGC retriever to KGC task, which can alleviate the long-tail problem without incurring additional training overhead. The proposed KICGPT model has a certain degree of innovation in part Knowledge Prompt, but the article lacks some methodological support or richer engineering experiments, and the experimental part has no way to verify the rationality of the motivation of this article, and the solution to the long tail problem is directly handed over to LLM. If so, it cannot be used as the motivation of this article in essence. Therefore, I think it is debatable whether this article needs to be accepted.

**Questions For The Authors:**

1. I don't quite agree with the long-tail problem of entity distribution mentioned in the article that the use of entity description information can solve the problem. In papers such as KG-BERT, there is no mention of the description that PLM can solve the long-tail problem. Is this motivation valid?
2. From the experimental results of WN18RR, the Text Self-Alignment part is relatively important, but the author's introduction to this part is too simple. How to obtain a fluent expression that is more in line with LLM, the author simply mentioned "The LLM is asked to summarize the semantics of the relation into a sentence".
3. It has been mentioned many times in the abstract and introduction that using LLM can solve the long tail problem, but there is no experimental support at all in the experimental part.
4. Generally speaking, the use of LLM is more suitable for few-shot tasks. It is recommended that the author consider the few-shot KGC task verification.

**Reasons To Accept:**

1. The article proposes to use LLM for KGC tasks, which breaks the paradigm of traditional KGC methods.
2. The author proposes to use Analogy Pool and Supplement Pool to reorder entities, which is innovative.
3. Experiments were carried out on the general FB15K-237 and WN18RR to verify the effectiveness of the proposed method.

**Reasons To Reject:**

1. The article lacks some methodological content, and more is to preprocess or carefully process the data, and the rationality of the corresponding processing method is difficult to weigh.
2. There is a gap between the motivation of the article and the final experiment. The content of the experimental part is not unified with the motivation. The most important thing is to deal with the long-tail problem. The experimental part should at least be supported by data.
3. The introduction of some key steps is not very clear, such as Text Self-Alignment, which feels to have a certain impact on the experimental results, but it is not introduced in detail.

**Reproducibility:**

3: Could reproduce the results with some difficulty. The settings of parameters are underspecified or subjectively determined; the training/evaluation data are not widely available.

**Reviewer Confidence:**

4: Quite sure. I tried to check the important points carefully. It's unlikely, though conceivable, that I missed something that should affect my ratings.

---

> ### Author Rebuttal · Authors · 2023-08-29
>
> Thanks for your constructive review! We address your concerns as follows.
>
>
> >  ${\bf Q1}$: I don’t quite agree with the long-tail problem
> of entity distribution mentioned in the article that the use of entity description
> information can solve the problem. In papers such as KG-BERT, there is no
> mention of the description that PLM can solve the long-tail problem. Is this
> motivation valid?
> There is a gap between the
> motivation of the article and the final experiment. The content of the experimental part is not unified with the motivation. The most important thing is
> to deal with the long-tail problem. The experimental part should at least be
> supported by data.
>  It has been mentioned many times in the abstract
> and introduction that using LLM can solve the long tail problem, but there is
> no experimental support at all in the experimental part.
>
> ${\bf Answer}$:
> The benefits of using PLM to process entity description information for long-tail entities have been observed in [1], which
> mentions that
> (line 21, right column, page 1)
> “Textual descriptions provide rich information for numerous semantic-related tasks, which brings an opportunity to learn informative representations for long-tail entities." In addition, we have provided experiments and analysis in Section 4.4 to evaluate the performance on long-tail entities. Specifically, Figure 3 shows that PLM-based approaches (such as StaR [2] and MEM-KGC [3]) have better performance on long-tail entities when compared with triple-based approaches (such as TransE and RotatE), and our approach has a more substantial performance improvement than the aforementioned approaches.
> Notably, our KICGPT method performs comparably or even better on long-tail entities (leftmost groups in Figure 3) compared to highly popular entities (rightmost groups in Figure 3) in the KGs.
>
> These results demonstrate the motivation and effectiveness of our proposed method for solving the long-tail problem.
>
> Moreover, we would like to emphasize that addressing the challenges posed by the long-tail distribution is only one of our contributions. Our approach also offers the following contributions:
>
> 1. As demonstrated in Table 1 and Figure 3, our KICGPT approach achieves overall state-of-the-art (SOTA) performance, and improves the performance not only on long-tail entities but also on almost all entities. This performance improvement is not limited to specific entity types or subsets, but rather across the entire Knowledge Graph.
>
> 2. Our work is the first to combine Language Models (LLMs) with triple-based Knowledge Graph Completion (KGC) methods
> to address
> the
> knowledge graph completion
> task.
>
> > ${\bf Q2}$: The introduction of some key steps is not very clear, such as Text Self-Alignment, which feels to have a certain impact on the experimental
> results, but it is not introduced in detail.
> From the experimental results of WN18RR, the Text Self-Alignment part is relatively important, but the author’s introduction to this part is too simple. How to obtain a fluent expression that is more in line with LLM, the author simply mentioned ”The LLM is asked to summarize the semantics of the relation into a sentence”.
>
> ${\bf Answer}$: Thank you for bringing this to our attention. As we are limited by page constraints, a full example is only shown in Appendix A.2, which illustrates how to perform text self-alignment by providing the entire prompt. In the revised version, we will provide more detailed descriptions of the text self-alignment process in the main text.
>
> In the following, we give the workflow of the text self-alignment. First, for relation $r$, we get its analogy pool (Section 3.3.1).  Then we order the analogy pool (Section 3.3.2). The ordered demonstrations are organized with prompt instructions that let the LLM find the relationship (refer to prompt in Appendix A.2). Finally, we use instruction prompts to let the LLM output only one concise sentence (section 3.4), which describes the relationship between the head entity (i.e., using placeholder [H] or 'B') and the tail entity (i.e., using placeholder [T] or 'A').
> By assigning real entities in triplets to placeholders, this sentence can be directly used as a description for the relation between entities.
>
>
> > ${\bf Q3}$:  Generally speaking,
> the use of LLM is more suitable for few-shot tasks. It is recommended that the
> author consider the few-shot KGC task verification.
>
>
> ${\bf Answer}$: The original KG has few triples for the long-tail entities. Thus, KGC for long-tail entities can be viewed as a few-shot task, in which each KG entity has only a few associated triples. Specifically, for each KG entity, a ''shot'' for that entity refers to its occurrence in a triple. Therefore, the number of degrees for each entity is the number of shots for that entity.
>
> In Section 4.4, we group entities based on the logarithm of their degrees (i.e., number of shots). The experimental results demonstrate that the proposed model performs well on few-shot cases (i.e., the leftmost groups in Figure 3).
>
>
> > ${\bf Q4}$:  The article lacks some methodological content, and more is to preprocess or carefully process the data, and the rationality of the corresponding processing method is difficult to weigh.
>
>
> ${\bf Answer}$: First, combining LLMs and KG structure is reasonable. The interaction between the knowledge in LLM parameters and the original KG leads to mutual alignment and enhancement. As can be seen from Table 1, KICGPT performs significantly better than ChatGPT and triple-based baselines. In this process, our in-context learning strategies also play an important role in bridging the KG structure with LLM reasoning.
>
> For the detailed design
> usefulness of the components
> (retriever, processing demonstrations, prompt engineering)
> can be seen from the ablation studies in Section 4.3.  Moreover, they are well-justified. For example, the significance of the initial order is also observed in the recent research on task reordering [4].
> Besides, simultaneously considering the diversity and correlation of demonstrations is important to ICL [5], which is similar to our construction of an analogy pool and a supplementary pool. We will add more discussions in the final revision.
>
> We will correct typos and polish the writing in the final revision.
>
> Sincerely thanks again for your patient and valuable review. If you have any other questions, please do not hesitate to contact us.
>
> Reference:
>
> [1] Language models as knowledge embeddings, IJCAI 2022.
>
> [2] Structure-augmented text representation learning for efficient knowledge graph completion, WWW 2021.
>
> [3] Mem-KGC: Masked entity model for
> knowledge graph completion with pre-trained language
> model, IEEE Access, 2021.
>
> [4] Is ChatGPT Good at Search?
> Investigating Large Language Models as Re-Ranking Agent, arxiv 2023.
>
> [5] Compositional Exemplars for In-context Learning, ICML 2023.

---

### Meta-Review · Area_Chair_4P8H · 2023-09-19

**Recommendation:** 4

**Metareview:**

The paper delves into the task of link prediction within incomplete KGs using an innovative technique that utilizes in-context learning (ICL) through a mechanism called Knowledge Prompt. They show that their method, KICGPT, demonstrates strong predictive abilities for missing entities in KGs. The authors claim that their approach is the first in combining Large Language Models (LLMs) with triple-based KGC methods for alleviating the long-tail problem in KGC.

The main concerns raised by the reviewers were:
1. Lack of clear motivation for the long tail problem - this was adequately addressed in the rebuttal as acknowledged by the reviewer
2. Use of a proprietary LLM (ChatGPT) - in response the authors have now repeated the experiments using Llama2 which is great

A few other minor issues were also adequately addressed by the authors.

Overall, I think the paper has some interesting ideas and a thorough evaluation. The community would benefit from these ideas and build upon them.

I request the authors to fit all the additional results provided during the rebuttal, in the main body of appendix of the paper, as appropriate.

---

### Decision · Program_Chairs · 2023-10-07

**Decision:**

Accept-Findings

**Comment:**

The paper delves into the task of link prediction within incomplete KGs using an innovative technique that utilizes in-context learning (ICL) through a mechanism called Knowledge Prompt. They show that their method, KICGPT, demonstrates strong predictive abilities for missing entities in KGs. The authors claim that their approach is the first in combining Large Language Models (LLMs) with triple-based KGC methods for alleviating the long-tail problem in KGC.

The main concerns raised by the reviewers were:
1. Lack of clear motivation for the long tail problem - this was adequately addressed in the rebuttal as acknowledged by the reviewer
2. Use of a proprietary LLM (ChatGPT) - in response the authors have now repeated the experiments using Llama2 which is great

A few other minor issues were also adequately addressed by the authors.

Overall, I think the paper has some interesting ideas and a thorough evaluation. The community would benefit from these ideas and build upon them.

I request the authors to fit all the additional results provided during the rebuttal, in the main body of appendix of the paper, as appropriate.